# Evaluation of Different Positive End-Expiratory Pressures Using Supreme™ Airway Laryngeal Mask during Minor Surgical Procedures in Children

**DOI:** 10.3390/medicina56100551

**Published:** 2020-10-21

**Authors:** Mascha O. Fiedler, Elisabeth Schätzle, Marius Contzen, Christian Gernoth, Christel Weiß, Thomas Walter, Tim Viergutz, Armin Kalenka

**Affiliations:** 1Clinic of Anesthesiology, Heidelberg University Hospital, 69120 Heidelberg, Germany; 2Clinic of Anesthesiology and Surgical Intensive Care Medicine, University Medical Centre Mannheim, 68167 Mannheim, Germany; eb.schaetzle@gmx.de (E.S.); tim.viergutz@umm.de (T.V.); 3Department of Anesthesiology and Intensive Care Medicine, Heilig-Geist-Hospital Bensheim, 64625 Bensheim, Germany; marius.contzen@artemed.de; 4Department of Anesthesiology, Surgical Intensive Care Medicine, Pain Therapy, Helios Hospital Duisburg, 47166 Duisburg, Germany; christian.gernoth@helios-gesundheit.de; 5Department of Medical Statistics, University Medical Centre Mannheim, 68167 Mannheim, Germany; christel.weiss@medma.uni-heidelberg.de; 6Emergency Department, University Medical Centre Mannheim, 68167 Mannheim, Germany; thomas.walter.med@umm.de; 7Department of Anesthesiology and Intensive Care Medicine, Hospital Bergstrasse, 64646 Heppenheim, Germany; armin.kalenka@kkh-bergstrasse.de

**Keywords:** paediatric anaesthesia, laryngeal mask, gastric insufflation, PEEP, airway devices, respiratory function

## Abstract

*Background and objectives:* The laryngeal mask is the method of choice for airway management in children during minor surgical procedures. There is a paucity of data regarding optimal management of mechanical ventilation in these patients. The Supreme™ airway laryngeal mask offers the option to insert a gastric tube to empty the stomach contents of air and/or gastric juice. The aim of this investigation was to evaluate the impact of positive end-expiratory positive pressure (PEEP) levels on ventilation parameters and gastric air insufflation during general anesthesia in children using pressure-controlled ventilation with laryngeal mask. *Materials and Methods:* An observational trial was carried out in 67 children aged between 1 and 11 years. PEEP levels of 0, 3 and 5 mbar were tested for 5 min in each patient during surgery and compared with ventilation parameters (dynamic compliance (mL/cmH_2_O), etCO_2_ (mmHg), peak pressure (mbar), tidal volume (mL), respiratory rate (per minute), FiO_2_ and gastric air (mL)) were measured at each PEEP. Air was aspirated from the stomach at the start of the sequence of measurements and at the end. *Results*: Significant differences were observed for the ventilation parameters: dynamic compliance (PEEP 5 vs. PEEP 3: *p* < 0.0001, PEEP 5 vs. PEEP 0: *p* < 0.0001, PEEP 3 vs. PEEP 0: *p* < 0.0001), peak pressure (PEEP 5 vs. PEEP 3: *p* < 0.0001, PEEP 5 vs. PEEP 0: *p* < 0.0001, PEEP 3 vs. PEEP 0: *p* < 0.0001) and tidal volume (PEEP 5 vs. PEEP 3: *p* = 0.0048, PEEP 5 vs. PEEP 0: *p* < 0.0001, PEEP 3 vs. PEEP 0: *p* < 0.0001). All parameters increased significantly with higher PEEP, with the exception of etCO_2_ (significant decrease) and respiratory rate (no significant difference). We also showed different values for air quantity in the comparisons between the different PEEP levels (PEEP 5: 2.8 ± 3.9 mL, PEEP 3: 1.8 ± 3.0 mL; PEEP 0: 1.6 ± 2.3 mL) with significant differences between PEEP 5 and PEEP 3 (*p* = 0.0269) and PEEP 5 and PEEP 0 (*p* = 0.0209). *Conclusions*: Our data suggest that ventilation with a PEEP of 5 mbar might be more lung protective in children using the Supreme™ airway laryngeal mask, although gastric air insufflation increased with higher PEEP. We recommend the use of a laryngeal mask with the option of inserting a gastric tube to evacuate potential gastric air.

## 1. Introduction

The laryngeal mask is a well-established option for airway management in pediatric patients undergoing general anesthesia for various surgeries [1]. This device can be used to secure ventilation in difficult situations, for example, after primary failure of endotracheal intubation [2,3,4].

Laryngeal mask airway (LMA) provides a relatively safe airway for positive pressure ventilation (PPV) in children [5]. Pressure control ventilation (PCV) is widely discussed as the method of choice for delivery of PPV through an LMA. A study by Natalini et al. compared pressure-controlled ventilation and volume-controlled ventilation with the LMA. The study demonstrated that the use of PCV during general anesthesia with the LMA reduced the peak airway pressure compared with volume control ventilation at the same tidal volumes and inspiratory times [6]. Positive end-expiratory pressure (PEEP) is frequently used in tracheally intubated patients to increase oxygenation, but is rarely used with the LMA because the low pressure seal predisposes to oropharyngeal and esophageal air leaks [7].

The use of general anesthetic reduces functional residual capacity especially in children, resulting in increased intrapulmonary shunts [8]. PEEP reduces this shunt volume during controlled ventilation in patients with healthy lungs [9]. There are no guidelines for PEEP settings in pediatric patients. Nevertheless, anesthesiologists traditionally set PEEP to a lower level in pediatric patients than in adults, i.e., below 5 cmH_2_O [8].

Optimization of functional residual capacity is even more important in children since they have a lower capacity for elastic retraction and a lower relaxation volume, and as a result are more susceptible to atelectasis than adults [10]. However, there is little data on optimum ventilation using a laryngeal mask and applying PEEP.

The Supreme™ laryngeal mask (S-LMA, a second-generation laryngeal mask) offers the option of simultaneous insertion of a gastric tube. This is important as, in contrast to airway management using endotracheal intubation, use of a laryngeal mask is potentially associated with the risk of gastric air insufflation with possible further consequences. But a randomized controlled trial by Drake-Brockman et al. evaluated the effect of endotracheal tubes versus LMAs on perioperative respiratory adverse events (PRAE) in infants [11]. The primary outcome of this study was the incidence of any PRAE in relation to the type of airway device used. The impact of LMA vs. endotracheal tubes (ETT) on the incidence of individual PRAE and their timing (intraoperatively and postoperatively) were assessed as secondary outcomes. This study showed a clear benefit of the use of an LMA compared with an endotracheal tube in a large number of infants undergoing minor elective surgery.

The aim of our observational investigation was to evaluate the effects on ventilation parameters during general anesthesia in children using pressure-controlled ventilation with the S-LMA at different PEEP levels. Primary outcome parameters were changes of the dynamic compliance and end-tidal carbon dioxide (etCO_2_) to verify recruitment maneuver’s with PEEP in lungs; secondary outcome parameters were the gastric air insufflation during ventilation with three different PEEP levels.

## 2. Methods

This prospective clinical trial was carried out from February 2012 to August 2014. Included children were aged between 1 and 11 years old, with American Society of Anaesthesiologists (ASA) classification I-III, scheduled for a minor elective surgery (inguinal hernia repair or circumcision) under general anesthesia in supine position with a planned surgery duration <30 min. Inclusion of patients was after informed consent. The study protocol was been approved by the Medical Ethics Committee II of the Mannheim Medical Faculty of the University of Heidelberg (2010-264N-MA; 22 June 2010).

Exclusion criteria were an ASA classification of IV and above, children with known difficult airways or impossibility of insertion of the laryngeal mask or the gastric tube. The size of the S-LMA (Teleflex Medical Europe Ltd., Athlone, Ireland) (see Table 1), the gastric tube and the cuff filling volume were selected based on weight-adapted tables provided by the manufacturer [12]. The cuff was inflated to a recommended maximum of 60 cmH_2_O using a cuff pressure monitor [12,13].

A total of 71 children were included in the study (ASA class I or II, all without lung disease). In two children investigation was stopped due to a leakage of the laryngeal mask and in two other children investigation had to be discontinued because the gastric tube could not be positioned for adequate function. These four children were excluded from the statistical analysis. Thus, the data for 67 children (12 girls and 55 boys) were available for final analysis.

### 2.1. General Anaesthesia

Each child was given premedication with midazolam (Dormicum^®^, Roche Pharma, Grenzach-Wyhlen, Germany) (0.5 mg/kg bodyweight (bw)) per os within 30 min of induction of anesthesia.

Balanced or total intravenous anesthesia was used. Each patient was connected to a Dräger Primus^®^ (Drägerwerk, Lübeck, Germany) machine. Standard monitors included precordial stethoscope, pulse oximeter, electrocardiography, automated noninvasive blood pressure (NIBP), capnometer and nasopharyngeal temperature.

The subject was pre-oxygenated with an inspiratory oxygen fraction of 80% and 4 L fresh gas flow per minute using a facemask. Thereafter we administered 2–4 µg fentanyl (Fentanyl Janssen^®^, Jansen-Cilag, Neuss, Germany) per kilogram bw and 4–6 mg propofol 10 mg/mL (Propofol^®^, Fresenius Kabi, Bad Homburg, Germany) per kilogram bw via a previously inserted intravenous cannula. Muscle relaxants were not required at any point during the investigation.

We did not perform bag-valve mask ventilation. The subject was pre-oxygenated and after the administration of fentanyl and propofol we inserted the S-LMA. At this time point, no monitoring of gastric air insufflation was undertaken. Lidocaine gel (Xylocain Gel 2%, Astra Zeneca, Wedel, Germany) recommended as a lubricant by the manufacturer [12], was applied to the back of the S-LMA prior to insertion. Adequate ventilation was verified based on the gel displacement test [12,14], bilaterally visible respiratory excursion, bilateral auscultation of the breath sounds and capnography.

A leakage test was carried out after the laryngeal mask had been fixed in place and connected to the ventilator. This was the airway pressure generated when an audible noise was heard over the mouth. Airway leak pressure was measured beforehand at a minimum pressure of 18 cmH_2_O and a maximum pressure of 25 cmH_2_O. We excluded patients from our study if the airway leak pressure was under the minimum pressure.

Pressure-controlled ventilation was carried out with a tidal volume of 6 to 8 mL/kg bw and a PEEP of 3 mbar. The inspiratory oxygen fraction (F_i_O_2_) was reduced from 80% to 50%, aiming at an oxygen saturation of above 95% and an end-tidal carbon dioxide (etCO_2_) concentration of 33–39 mmHg. A balanced anesthesia was maintained to the end of the surgery, using sevofluran (Sevofluran Baxter, Baxter, Unterschleißheim, Germany) with a minimal alveolar concentration (MAC) of 0.8, or as a total intravenous anesthesia using propofol. The equilibration period was not defined.

After positioning of the S-LMA, each patient received a lubricated gastric tube through the drainage canal. Measurement time started after the insertion of the gastric tube and the withdrawal of the contents of the stomach (fluids or air) with a 5 mL syringe.

### 2.2. Data Collection

The recorded ventilation parameters included the dynamic compliance, etCO_2_, peak inspiratory pressure (PIP), inspiration volume, respiratory rate (RR), and the FiO_2_ for each PEEP level. Additionally, gastric air and aspirates/secretions were documented. We did not measure the BMI of the children and we didn’t take the time of the surgery during our measurements.

Once anesthesia was established, PEEP was increased to 5 mbar for 5 min (T0), PEEP was then reduced to 3 mbar for 5 min (T1) and to 0 mbar for 5 min (T2). Peak inspiration pressure was kept constant during all PEEP levels. All measurements were performed during the surgical procedure. The patients were not breathing spontaneously. The laryngeal mask was removed correctly at the end of surgery once the patient was awake and exhibiting sufficient spontaneous respiration and an adequate presence of protective reflexes.

### 2.3. Statistical Analysis

Statistical analysis was carried out using the statistical software package SAS, release 9.4 (SAS Institute, Cary, NC, USA).

Quantitative variables are presented as mean and standard deviation together with their range (see Table 1). Data approximately normally distributed (i.e., dynamic compliance, etCO_2_, peak pressure, tidal volume, respiratory rate) that had been recorded multiple times for a given observation unit was analyzed using repeated measures ANOVA. The SAS procedure PROC MIXED with the fixed factors “measuring point”, patients’ age and body weight group and the random factor “patient ID” was used for this purpose based on three defined PEEP levels (T0: PEEP 5, T1: PEEP 3 and T2: PEEP 0 mbar).

To compare gastric air at different time points the Friedman test was used, since this parameter may not be considered normally distributed. For pairwise comparisons of measurement time points, post hoc tests according to Scheffé or Wilcoxon test for 2 paired samples were used, respectively. The result of a statistical test was considered as significant for *p* < 0.05.

## 3. Results

Baseline characteristics are presented in Table 1. The youngest child was aged 13 months and the oldest 10 years and 10 months. Mean body weight was 19.5 ± 8.4 kg.

### 3.1. Ventilation Parameters

Comparison of RR, tidal volume (V_t_), peak pressure, dynamic compliance, expiratory carbon dioxide concentration (etCO_2_), inspiratory oxygen fraction (FiO_2_) and quantity of gastric air at T0 (PEEP 5 mbar), T1 (PEEP 3 mbar) and T2 (PEEP 0 mbar) are shown in Table 2.

For each PEEP level—with the only exception of respiratory rate, *p* = 0.3708—changes over time could be detected (each *p* < 0.0001) (Table 2).

There was a significant decrease in dynamic compliance as PEEP levels reduced (PEEP 5 > PEEP 3 > PEEP 0).

A significant increase in etCO_2_ concentration was observed with decreasing PEEP (PEEP 5 < PEEP 3 < PEEP 0). The inspiratory oxygen fraction (FiO_2_) was not significant.

The mean airway leak pressure was 22.5 ± 2 cmH_2_O.

### 3.2. Gastric Air or Aspirates/Secretion

An important endpoint of our study was the amount of gastric air or aspirates during mechanical ventilation with S-LMA and three different PEEP levels.

Also, for this parameter, changes over time have been found (*p* = 0.0176). Wilcoxon tests for two paired samples revealed significant differences between PEEP 5 and PEEP 3 (*p* = 0.0269) as well as PEEP 5 and PEEP 0 (*p* = 0.0209). We did not aspirate any gastric secretion.

## 4. Discussion

The principal findings of the present observational clinical trial are that a PEEP of 5 mbar provides significantly higher dynamic compliance (C_dyn_), tidal volume (V_t_) and peak pressure during general anesthesia in children using PCV with the S-LMA at different PEEP levels. We also found a significantly higher gastric air volume during ventilation with a PEEP of 5 mbar.

The S-LMA with the additional option of insertion of a gastric tube was used in this investigation of different PEEP levels during minor elective surgical interventions in children. We used this S-LMA because with the opportunity of insertion of a gastric tube during PCV and the application with PEEP, the insufflation of air and the risk of aspiration seems to be higher.

According to several studies, sufficient PEEP should be used to minimize atelectasis and maintain oxygenation [5,9,15,16]. Serafini et al. examined ten children, ranged from ages 1 to 3 years, all without lung disease. After general anesthesia for cranial or abdominal CT scans, pulmonary morphology was investigated. A PEEP of 5 cmH_2_O was shown to recruit all available alveolar units and to induce the disappearance of atelectasis in dependent lung regions [17]. However, after full muscle relaxation, ventilation was with an orotracheal tube and not with a laryngeal mask in this study. Our study demonstrated that without muscle relaxation the children might develop atelectasis, and that ventilation with S-LMA and PEEP of 5 mbar improves the dynamic compliance and recruited the lungs.

Goldmann and colleagues tested the hypothesis that in anaesthetized pediatric patients the ProSeal™ laryngeal mask (P-LMA) can be used effectively to apply a PEEP of 5 cmH_2_O during pressure-controlled ventilation (PCV) and that this leads to an improved arterial oxygenation compared to a PCV ventilation without PEEP [18]. It seems that the application of PEEP (5 cmH_2_O) during PCV improves gas exchange in healthy pediatric patients [18]. We did not take arterial blood gas samples in our setting. The duration of minor surgeries in our setting was not as long as the procedures in the study from Goldman et al.

A randomized controlled trial with 90 children showed that PCV with PEEP using the P-LMA was accompanied with lower incidence of adverse events in comparison to spontaneous respiration in infants and toddlers with upper respiratory tract infection undergoing infra umbilical surgeries under general anesthesia. The authors concluded that PCV with PEEP using P-LMA may be the preferred mode of ventilation in children [19].

In our study, we found significant differences in C_dyn_ through different PEEP levels during PCV. C_dyn_ was significantly greater for a PEEP of 5 mbar. In pediatric patients PEEP is traditionally set lower, but we have not found profound reasoning in the literature, empirically, anesthesiologists tend to ventilate children with a lower PEEP compared to adult patients. Wirth et al. investigated whether moderately higher PEEP improves respiratory mechanics and regional ventilation. Therefore, 40 children were mechanically ventilated with PEEP 2 and 5 cmH_2_O. They analyzed volume-dependent compliance profiles as a measure of intratidal recruitment/derecruitment. They concluded that increasing PEEP from 2 to 5 cmH_2_O improved mean compliance and was associated with improved peripheral ventilation without causing overdistension of the lungs or hemodynamic compromise [8]. This was the first study investigating the effects of PEEP on intratidal compliance in children. Compared to our study children received full neuromuscular block, they did a tracheal intubation during anesthesia and invasive ventilation and the planned surgery duration was >60 min.

In a study by von Ungern-Sternberg et al. [10], younger children were more susceptible to atelectasis than older children and benefited from higher PEEP settings. Another study in children showed that increasing the PEEP can reopen dorsal areas of the lungs [17]. As the closing capacity is lower at younger age, younger children have a higher probability that closing capacity is lower than FRC (functional residual capacity). As a consequence, small airways tend to collapse at the end of expiration. Therefore, particularly in younger children a higher PEEP might be required to shift FRC to a level at which the collapse of the small airways is prevented [8,10].

Furthermore, we demonstrated in our study that the behavior of etCO_2_ concentration opposed that of C_dyn_. EtCO_2_ concentration was significantly higher at a PEEP of 0 mbar than at 5 mbar. This is possibly due to the higher C_dyn_ arising from the larger gas exchange area (recruitment with PEEP and minute volume), whereby the carbon dioxide is more easily exhaled.

In our study, we posed the question of whether increasing the PEEP level results in an increase in the rate of gastric air insufflation during ventilation with S-LMA. The analyses revealed that there were significant differences in the quantities of gastric air obtained via the gastric tube for the different PEEP levels in the overall data analysis. Lagarde et al. noted in their study that the incidence of gastric air insufflation rises with increasing inspiratory pressure under face mask ventilation [20]. The ventilation with a face mask for further pre-oxygenation prior to positioning of the S-LMA is probably the reason for positive quantities of air. Therefore, we did not ventilate each child during pre-oxygenation with the mask.

We also recorded in our study the highest peak pressure with a PEEP of 5 mbar merely around 15 mbar and significant reduction of the gastric air quantity with the decrease of the PEEP levels. Lagarde et al. describe gastric air insufflation in children as occurring in over 58% at an inspiratory pressure of above 15 cmH_2_O [20]. However, gastric air insufflation was detected through auscultation of the stomach in this experimental setup. How high the rates for false positives and negatives were in this study is unclear. According to the literature, gastric air insufflation occurs at an earlier stage in younger subjects than in older ones. A limit of below 15 cmH_2_O for inspiratory pressure and, in some cases even below 10 cmH_2_O, is referred for children below the age of one year. The inspiratory pressure therefore appears to be age-dependent [20]. Bouvet et al. reach the conclusion that an inspiratory pressure of 15 cmH_2_O is probably the best compromise between adequate ventilation with a face mask and gastric air insufflation [21]. However, PEEP is not used in this study and the reduced incidence of gastric air insufflation is related to the induction phase of anesthesia [21]. The probability of gastric air insufflation arises due to the use of PCV, as this results in lower inspiratory pressures for volume-controlled ventilation at the same tidal volume. The inspiratory pressure of 15 cmH_2_O is recommended as the standard limit in children as no further increase in tidal volume is achieved and there is an increased incidence of gastric air insufflation above this value [20]. The correct positioning of the airway device, such as the laryngeal mask, appears to be the decisive factor in achieving optimum sealing and avoiding any potential axial rotation, thereby ensuring that less air enters the stomach [22,23].

A further question that arises repeatedly is whether insufflation of air increases the patients’ risk of aspiration.

The results of our investigation showed a tendency that the quantity of the gastric air can be reduced following aspiration through the gastric tube in the S-LMA. Maybe the opportunity of a gastric tube can lower the risk of regurgitation and aspiration.

However, the precise reduction in the risk of aspiration cannot be derived from our available data. In a retrospective analysis, Bernadini et al. demonstrated that, compared to an endotracheal tube, there is no increase in the risk of aspiration when using a laryngeal mask, but that the majority of cases of aspiration occurred in patients who required emergency surgery [24].

Our investigation was subject to some limitations. The patients included in this study were aged between 1 year and 11 years. The range of age is very large and consequently the group is heterogenous (weight and height). Therefore, the results obtained in the present investigation cannot be directly extrapolated to younger children and certainly not to older children or even adults. To minimize the risk of complications, only children with uncomplicated airways were investigated. Furthermore, this is not a blinded study as no experimental and control groups were formed. A critical view must also be taken in relation to the short duration of 5 min for the application of PEEP. It might be possible to obtain more clear-cut results if both the individual PEEPs were tested for longer periods and a greater overall number of subjects with a better age distribution were to be investigated. The range of different PEEP levels were very small. In our study, the focus was set on these levels because pediatric anesthetists are used to lower PEEP levels.

We also used only one kind of laryngeal mask with the advantage of a channel for a gastric tube. Additionally, the amount of gastric air should better be evaluated by gastric ultrasound or auscultation, a method well established in anesthesia practice today. The evacuation of gastric air over the gastric tube by aspiration via a syringe is not as valid as gastric ultrasound. This is a limitation of our study and we need further randomized investigations with more patients to figure out the risk of aspiration during ventilation with laryngeal masks.

## 5. Conclusions

Our results revealed that a higher PEEP, maximum 5 mbar in our investigation, yielded more ventilator-associated advantages than disadvantages with the S-LMA. Results on this were usually significant, especially for a PEEP of 5 mbar, with a larger inspiration volume, greater C_dyn_ and a lower etCO_2_ concentration.

Our investigation also demonstrates that significant quantities of air are insufflate into the stomach under PCV with the S-LMA and a PEEP of up to 5 mbar.

However, it must be noted, that air was collected mainly after induction of anesthesia and ventilation with the S-LMA after insertion of this device. The question therefore arises, as to whether routine aspiration of air from the stomach significantly reduces the incidence of aspiration when the S-LMA, and possibly also other supraglottic airway devices, are used. This could constitute a hypothesis for future studies with a larger sample size.

Overall, the conclusion can at least be drawn that a positive PEEP value is more suitable than no PEEP during ventilation with S-LMA in children.

## Figures and Tables

**Table 1 medicina-56-00551-t001:** Baseline characteristics.

Characteristic			All Patients (*n* = 67)
**Age, year**			4.7 ± 2.4
Range			1.1–10.8
**Gender**	***n* (%)**		
Male			55 (82)
Female			12 (18)
**Body weight *, kg**			
			19.5 ± 8.4
Range, kg			9–59
**Groups ***	***n* (%)**		
	<5 kg	Supreme™ laryngeal mask size (1)	0 (0)
	5–10 kg	Supreme™ laryngeal mask size (1.5)-	5 (8)
	10–20 kg	Supreme™ laryngeal mask size (2)	38 (58)
	20–30 kg	Supreme™ laryngeal mask size (2.5)	17 (26)
	30–50 kg	Supreme™ laryngeal mask size (3)	5 (8)
	50–70 kg	Supreme™ laryngeal mask size (4)	1 (2)

***** Information on body weight was unavailable for one subject.

**Table 2 medicina-56-00551-t002:** Ventilation parameters and gastric air by PEEP.

	T0 (PEEP 5) (mbar)	T1 (PEEP 3) (mbar)	T2 (PEEP 0) (mbar)	*p*-Values for Pairwise Comparisons
**Dynamic compliance C_dyn_** **(mL/cmH_2_O)**	18.4 ± 7.5	16.8 ± 6.9	14.4 ± 5.5	T0-T1: <0.0001 T0-T2: <0.0001 T1-T2: <0.0001
**etCO_2_ (mmHg)**	37.1 ± 4.7	38.2 ± 4.3	41.3 ± 5.1	T0-T2: <0.0001 T1-T2: <0.0001
**Peak pressure** **(mbar)**	14.9 ± 1.6	13.0 ± 1.6	10.6 ± 1.5	T0-T1: <0.0001 T0-T2: <0.0001 T1-T2: <0.0001
**V_t_ (mL)**	170.4 ± 66.2	160.2 ± 60.8	138.8 ± 50.2	T0-T1: 0.0048 T0-T2: <0.0001 T1-T2: <0.0001
**RR** **(per minute)**	20.9 ± 3.8	20.9 ± 3.8	21.0 ± 3.8	not significant
**FiO_2_**	0.51 ± 0.06	0.49 ± 0.06	0.49 ± 0.06	not significant
**Gastric air (mL)**	2.8 ± 3.9	1.8 ± 3.0	1.6 ± 2.43	T0-T1: 0.0269 T0-T2: 0.0209

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
