# Peer review of "Evaluation of Different Positive End-Expiratory Pressures Using Supreme™ Airway Laryngeal Mask during Minor Surgical Procedures in Children"

_medicina, 2020, doi:10.3390/medicina56100551_

Round 1
Reviewer 1 Report
I have enjoyed reviewing your manuscript. This is a single center observational study on the use of different PEEP levels during ventilation with a supreme laryngeal mask in children. Although the topic would be of interest to anesthetists, the rationale, methods and discussion need much more work to bring it up to standard for publication.
As a summary I would like to have more data on type of operation, positioning. Can you provide more test subject characteristics? This may influence your outcome parameters.
What would you think would be the best primary outcome parameter showing that PEEP has a beneficial effect on gas exchange?
Furthermore, the position of the S-LMA is not verified by fiberoptic examination.
The amount of gastric air should be evaluated by gastric ultrasound, a method well established in anesthesia practice today.
Specific remarks:
Title: I feel as the title is a bit misleading, as you mention the study design and the primary outcome.
Abstract:
L22: In the Background section you state that one can empty the stomach content of aspirate. The stomach does not contain aspirate, rather air and gastric juice.
LL24: Please define primary and secondary outcome parameters
LL25: this is rather an observational trial
LL35: Referring to changes of the respiratory rate: were the patients breathing spontaneously?
LL39: If you suggest an advantage, why do you not state what kind of advantage?
Introduction:
LL 68: Please provide a rationale why you used a second generation supraglottic airway device.
LL 75: Please specify clear primary and secondary outcome parameters
Methods:
LL 89: A cuff pressure of 60 cmH2O may cause oropharyngeal discomfort postoperatively. Please provide more information on this.
LL110: How long the children received bag-valve mask ventilation? At what pressure limits? Was there a monitoring of gastric air insufflation at this time point?
LL 113: Reference #12 is not available: Please provide an available reference. How was the leakage test performed? Did you perform a leak pressure test before you inserted the gastric tube?
LL114: How did you confirm correct placement of the S-LMA?
LL 139: How did you get your sample size? Did you perform a power calculation?
Results:
LL153: What types of surgery were done? How was the positioning of the patients. What was the BMI? LL165:How did you perform the mean airway leak pressure?
Please provide data on oxygenation.
Discussion:
The discussion section requires major reworking.
LL179: What exactly do you aim at with this sentence?
LL181-186: This children were not only tracheally intubated but received also full muscle relaxation. This may be a primary reason for atelectasis.
LL187: In this study, arterial oxygenation was the primary outcome parameter.
LL200: In this study children received full neuromuscular block.
LL217: You state, that an increased gas exchange area was the reason for the lower etCO2. I doubt this conclusion. When you multiply the respiratory rate with the tidal volume you will receive the minute volume, thereby decarboxylation increases.
LL219: The evacuation of gastric air by a gastric tube seems to be a method, not as valid as gastric ultrasound or even gastric auscultation. Please provide a reference for this method and their validity. Please discuss the limitation of your methodology.
Furthermore I doubt the clinical relevance, albeit statistical significant, of 0.5 ml or 1 ml gastric air. By bag-valve-mask ventilation prior to inserting the S-LMA, you may have insufflated much more air than by S-LMA ventilation during the study sequences.
LL242: How did you confirm correct S-LMA positioning?
LL 247: I cannot follow your conclusion that, based on your data,the risk of regurgitation or aspiration was reduced.
Conclusions:
LL 270: You mix conclusion with limitations. The face mask ventilation prior to your study sequence is a major drawback of this study.
Author Response
October 7, 2020
Dr. Mascha O. Fiedler, MD
Editor-in-Chief, Journal Medicina
Dear Professor Dr. Stankevičius,
On behalf of my coauthors, I would like to thank you for the opportunity to revise and resubmit our manuscript medicina-957030, entitled “Evaluation of different positive end-expiratory pressures using Supreme™ Airway laryngeal mask during minor surgical procedures in children.”
We found the reviewers’ comments to be helpful in revising the manuscript and have carefully considered and responded to each suggestion. In the majority of cases we were successful in incorporating the reviewers’ feedback into our revised manuscript.
We have included a response to reviewers in which we address each comment the reviewers made. In our response to reviewers, the reviewers’ comments are numbered, and our responses follow below, in blue, and are prefaced by “Author response.” Corresponding changes are highlighted in the manuscript text in the revised file.
Thank you again for your consideration of our revised manuscript.
Sincerely,
Dr. Mascha O. Fiedler, MD
Im Neuenheimer Feld 420
D-69120 Heidelberg
Germany
mascha.fiedler@med.uni-heidelberg.de
Comments from Reviewer 1:
- Comment 1: I would like to have more data on type of operation, positioning. Can you provide more test subject characteristics? This may influence your outcome parameters.
Response 1: Thank you for pointing this out. Types of surgery were minor elective surgery (inguinal hernia repair or circumcision) with a planned surgery duration <30 min. The positioning of the patients was supine. Unfortunately, we are not able to provide more test subject characteristics. We add this suggestion LL 133-136.
- Comment 2: What would you think would be the best primary outcome parameter showing that PEEP has a beneficial effect on gas exchange?
Response 2: We believe that the best primary outcome parameters in this setting are dynamic compliance and expiratory carbon dioxide concentration (etCO2) because of the recruitment maneuver’s with PEEP in lungs. (LL 83-86)
- Comment 3: Furthermore, the position of the S-LMA is not verified by fiberoptic examination.
Response 3: Thank you for this suggestion. In our opinion verifying the position of the S-LMA by fiberoptic examination is not daily practice in anesthesia during minor surgical procedures in children.
- Comment 4: The amount of gastric air should be evaluated by gastric ultrasound, a method well established in anesthesia practice today.
- Response 4: We agree with this and have incorporated your suggestion throughout the manuscript. LL 310-314
Specific remarks from Reviewer 1:
- Title: I feel as the title is a bit misleading, as you mention the study design and the primary outcome.
Response: Thank you for this comment. We think the study design is mentioned in the title and this might be interesting for readers.
Abstract:
- L22: In theBackground section you state that one can empty the stomach content of aspirate. The stomach does not contain aspirate, rather air and gastric juice.
Response: We agree with this and have incorporated your suggestion throughout the manuscript. LL 22
- LL24: Please define primary and secondary outcome parameters
Response: Primary outcome parameters were changes of the dynamic compliance and end-tidal carbon dioxide (etCO2); secondary outcome parameters were the gastric air insufflation during ventilation with three different PEEP levels. LL 83
- LL25: this is rather an observational trial
Response: We agree with this and have incorporated your suggestion throughout the manuscript. LL 25
- LL35: Referring to changes of the respiratory rate: were the patients breathing spontaneously?
Response: The patients were not breathing spontaneously. LL156
- LL39: If you suggest an advantage, why do you not state what kind of advantage?
Response: The advantage with the PEEP of 5mbar seems to be more lung protective. LL39
Introduction:
- LL 68: Please provide a rationale why you used a second generation supraglottic airway device.
Response: We used a second generation supraglottic airway device because it offers the option of simultaneous insertion of a gastric tube. LL 73
- LL 75: Please specify clear primary and secondary outcome parameters
Response: In the cited study the primary outcome was the incidence of any PRAE in relation to the type of airway device used. The impact of LMA vs. ETT on the incidence of individual PRAE and their timing (intraoperatively and postoperatively) was assessed as secondary outcomes.
Our outcome parameters were recommended in the answer to LL 24.
Methods:
- LL 89: A cuff pressure of 60 cmH2O may cause oropharyngeal discomfort postoperatively. Please provide more information on this.
Response: In the Post Awakening Care Unit (PACU) we did not investigate oropharyngeal discomfort.
- LL110: How long the children received bag-valve mask ventilation? At what pressure limits? Was there a monitoring of gastric air insufflation at this time point?
Response: In this setting we did not perform bag-valve mask ventilation. The subject was pre-oxygenated and after the administration of fentanyl and propofol we inserted the S-LMA. At this time point was no monitoring of gastric air insufflation. LL129
- LL 113: Reference #12 is not available: Please provide an available reference. How was the leakage test performed? Did you perform a leak pressure test before you inserted the gastric tube?
Response: The available reference is now: https://www.lmaco-ifu.com/sites/default/files/node/438/ifu/revision/3593/ifu-lma-supreme-paj2100002buk.pdf
We changed it in our manuscript.
A leakage test was carried out after the laryngeal mask had been fixed in place and connected to the ventilator. This was the airway pressure generated when an audible noise was heard over the mouth. Airway leak pressure was measured beforehand at a minimum pressure of 18 cmH2O and a maximum pressure of 25 cmH2O. We excluded patients from our study if the airway leak pressure was under the minimum pressure. LL 135
After positioning of the S-LMA each patient received a lubricated gastric tube through the drainage canal. LL150
- LL114: How did you confirm correct placement of the S-LMA?
Response: As described with the gel displacement test. Water-soluble gel (0.5-1 ml) is placed at the proximal end of the drain tube so that it forms a column of about 2-3 cm. Minimal movement or gentle up and down movements indicates a normal position.
Sharma B, Sood J, Sahai C, Kumra VP. Troubleshooting ProSeal LMA. Indian J Anaesth. 2009;53(4):414-24.
- LL 139: How did you get your sample size? Did you perform a power calculation?
Response: We did not do a power calculation before we began our single-center observational study. We did not find enough data in the literature even in the setting ventilation with LMA and PEEP in children at the start of our study.
Results:
- LL153: What types of surgery were done? How was the positioning of the patients? What was the BMI?
Response: Types of surgery were minor elective surgery (inguinal hernia repair or circumcision). The positioning of the patients was supine. Unfortunately, we didn’t measure the BMI of the children and we didn’t take the time of the surgery during our measurements. LL 156
- LL165: How did you perform the mean airway leak pressure?
Response: see the answer to LL 113 to leakage test.
- Please provide data on oxygenation.
Response: Thank you for this suggestion. We provide data on oxygenation. But during the ventilation with three different PEEP levels the inspiratory oxygen fraction (FiO2) was not significant. We did not take any blood gas samples. The SpO2 was > 95%.
Discussion:
The discussion section requires major reworking.
- LL179: What exactly do you aim at with this sentence?
Response: We used this S-LMA because with the opportunity of insertion of a gastric tube during PCV and the application with PEEP the insufflation of air and the risk of aspiration seems to be higher. LL208
- LL181-186: This children were not only tracheally intubated but received also full muscle relaxation. This may be a primary reason for atelectasis.
Response: We agree with your statement. Our study demonstrated that without muscle relaxation the children developed atelectasis and ventilation with S-LMA and PEEP of 5mbar improves the dynamic compliance and recruited the lungs. LL217
- LL187: In this study, arterial oxygenation was the primary outcome parameter.
Response: We did not take arterial blood gas sample in our setting. The duration of minor surgeries in our setting was not as long as the procedures in the study from Goldman et al. LL 228
- LL200: In this study children received full neuromuscular block.
Response: We agree with you. The children in this study underwent elective ENT surgery with a planned surgery duration >60 min. LL 246
- LL217: You state, that an increased gas exchange area was the reason for the lower etCO2. I doubt this conclusion. When you multiply the respiratory rate with the tidal volume you will receive the minute volume, thereby decarboxylation increases.
Response: We agree with your opinion. The RR in our group was nearly the same. The tidal volume during the ventilation was higher with a PEEP of 5mbar. So, the reason for the lower etCO2 is an increased gas exchange area as a result of recruiting with PEEP and minute volume during PCV. LL 255
- LL219: The evacuation of gastric air by a gastric tube seems to be a method, not as valid as gastric ultrasound or even gastric auscultation. Please provide a reference for this method and their validity. Please discuss the limitation of your methodology.
Response: Thank you for pointing this out. The evacuation of gastric air by gastric tube is not as valid as gastric ultrasound. This is a limitation of our study and we need further randomized investigations with more patients to figure the risk of aspiration during ventilation with laryngeal masks out. LL 314
- Furthermore I doubt the clinical relevance, albeit statistical significant, of 0.5 ml or 1 ml gastric air. By bag-valve-mask ventilation prior to inserting the S-LMA, you may have insufflated much more air than by S-LMA ventilation during the study sequences.
Response: We are sorry with this misleading information about the ventilation with a mask prior to inserting the S-LMA. To avoid the insufflation of air we did not ventilate each child with a mask. We only put a mask around nose and mouth for pre-oxygenation as we described before in the method part.
- LL242: How did you confirm correct S-LMA positioning?
Response: We insert the mask as described in the instruction for S-LMA and we confirm the correct positioning with the leakage test as described before.
http://www.lmaco-ifu.com/sites/default/files/node/438/ifu/revision/3593/ifu-lma-supreme-paj2100002buk.pdf.
- LL 247: I cannot follow your conclusion that, based on your data, the risk of regurgitation or aspiration was reduced.
Response: The results of our investigation showed a tendency that the quantity of the gastric air can be reduced following aspiration through the gastric tube in the S-LMA. Maybe the opportunity of a gastric tube can lower the risk of regurgitation and aspiration. LL295
Conclusions:
- LL 270: You mix conclusion with limitations. The face mask ventilation prior to your study sequence is a major drawback of this study.
Response: We agree with this and have incorporated your suggestion throughout the manuscript.
Reviewer 2 Report
Dear Sirs
Thank you very much for the opportunity to review the manuscript „Evaluation of different positive end-expiratory pressures using Supreme Airway laryngeal mask during minor surgical procedures in children.
In this work, the authors compare the impact of different PEEP levels on respiratory related parameters during general anesthesia in children.
The topic is interesting and addresses a clinically relevant and equivocally discussed question.
However, the results of the presented study would hardly convince any pro or contra PEEP with LMA hardliner.
There are different reasons for this opinion:
- The study was done in 67 patients with a large range of age (and therefore weight and height). This is not enough to really proof safety.
- A possible remaining conclusion may be, PEEP improves respiratory dynamics even when using LMAs. This is not so surpizing, even though the differences in PEEP were small.
- The volume of air inflated in direction of the stomach instead of towards the lung depends on the muscular tone. Even though the patients were not paralyzed, the depth of anesthesia plays a role. There is no information about this and different modes of anesthetics were used.
- Patients received a gastric tube – how was this handled afterwards?
- The study periods were very short.
- What is the clinical reference of 2.8 vs. 1.8 ml of gastric air?
- How well introduced is the quantification of gastric air by aspiration via a syringe?
Minor topics:
- How was the mentioned leakage test performed?
Although most of these limitations are properly listed in the limitations section of the discussion, the limitations are still there…
Author Response
October 7, 2020
Dr. Mascha O. Fiedler, MD
Editor-in-Chief, Journal Medicina
Dear Professor Dr. Stankevičius,
On behalf of my coauthors, I would like to thank you for the opportunity to revise and resubmit our manuscript medicina-957030, entitled “Evaluation of different positive end-expiratory pressures using Supreme™ Airway laryngeal mask during minor surgical procedures in children.”
We found the reviewers’ comments to be helpful in revising the manuscript and have carefully considered and responded to each suggestion. In the majority of cases we were successful in incorporating the reviewers’ feedback into our revised manuscript.
We have included a response to reviewers in which we address each comment the reviewers made. In our response to reviewers, the reviewers’ comments are numbered, and our responses follow below, in blue, and are prefaced by “Author response.” Corresponding changes are highlighted in the manuscript text in the revised file.
Thank you again for your consideration of our revised manuscript.
Sincerely,
Dr. Mascha O. Fiedler, MD
Im Neuenheimer Feld 420
D-69120 Heidelberg
Germany
mascha.fiedler@med.uni-heidelberg.de
There are different reasons for this opinion:
- The study was done in 67 patients with a large range of age (and therefore weight and height). This is not enough to really proof safety.
Response: Thank you for pointing this out. We agree with you and mentioned this in our limitions. LL 303
- A possible remaining conclusion may be, PEEP improves respiratory dynamics even when using LMAs. This is not so surpizing, even though the differences in PEEP were small.
Response: We agree with your suggestion. We mentioned this in our discussion. LL 311
- The volume of air inflated in direction of the stomach instead of towards the lung depends on the muscular tone. Even though the patients were not paralyzed, the depth of anesthesia plays a role. There is no information about this and different modes of anesthetics were used.
Response: Thank you for pointing this out. The duration of the minor surgeries in our study wasn’t longer than 30 minutes. We kept this in mind and decided against the measuring of the depth of anesthesia (for example BIS monitoring). Therefore, it was interesting for us if the extubation period shows serious side effects (as aspiration or regurgitation).
- Patients received a gastric tube – how was this handled afterwards?
Response: The gastric tube inside of the S-LMA was removed completely at the end of anesthesia and sufficient spontaneous breathing. LL161
- The study periods were very short.
Response: We agree with this and mentioned this in our discussion.
- What is the clinical reference of 2.8 vs. 1.8 ml of gastric air?
Response: The study group is heterogenous (large age range) as we mentioned before and also in our limitations.
- How well introduced is the quantification of gastric air by aspiration via a syringe?
Response: The evacuation of gastric air by gastric tube is not as valid as gastric ultrasound. This is a limitation of our study and we need further randomized investigations with more patients to figure the risk of aspiration during ventilation with laryngeal masks out.
Minor topics:
- How was the mentioned leakage test performed?
Response: A leakage test was carried out after the laryngeal mask had been fixed in place and connected to the ventilator. This was the airway pressure generated when an audible noise was heard over the mouth. Airway leak pressure was measured beforehand at a minimum pressure of 18 cmH2O and a maximum pressure of 25 cmH2O. We excluded patients from our study if the airway leak pressure was under the minimum pressure.
Reviewer 3 Report
Thank you for submitting the manuscript. I have read your peper with great attention and interest. The topic you dealt with is certainly important in the field of pediatric anesthesia.
The laryngeal mask marked a turning point in the management of the airways, both for elective and urgent / emergency interventions.
Over the years, anesthesiologists have split into two factions: the proponents of the laryngeal mask and its detractors. I believe that your approach, scientific and critical, rightly places itself in the middle of this debate. Your paper is certainly well written, clear and essential.
All aspects of your research project have been addressed in a more than satisfactory manner. However, appreciate that you make minor revisions to your paper to increase its quality.
- First I would like you to better specify in the "methods" section how you have measured the gastric air content. This allows you to reproduce your study as faithfully as possible.
- I would like to see the individual contributions made by the authors in the section of the author's contribution. In fact from the draft it looks like you have pasted the template without compiling it.
- Likewise, I would like to see the section on funding correctly filled out
- Finally I believe that you must also correctly fill in the "Acknowledgments" section.
In conclusion, I believe that your paper is very interesting and that with the appropriate revisions it can be of high quality. I hope these comments are helpful to you.
Kind Regards
Author Response
October 7, 2020
Dr. Mascha O. Fiedler, MD
Editor-in-Chief, Journal Medicina
Dear Professor Dr. Stankevičius,
On behalf of my coauthors, I would like to thank you for the opportunity to revise and resubmit our manuscript medicina-957030, entitled “Evaluation of different positive end-expiratory pressures using Supreme™ Airway laryngeal mask during minor surgical procedures in children.”
We found the reviewers’ comments to be helpful in revising the manuscript and have carefully considered and responded to each suggestion. In the majority of cases we were successful in incorporating the reviewers’ feedback into our revised manuscript.
We have included a response to reviewers in which we address each comment the reviewers made. In our response to reviewers, the reviewers’ comments are numbered, and our responses follow below, in blue, and are prefaced by “Author response.” Corresponding changes are highlighted in the manuscript text in the revised file.
Thank you again for your consideration of our revised manuscript.
Sincerely,
Dr. Mascha O. Fiedler, MD
Im Neuenheimer Feld 420
D-69120 Heidelberg
Germany
mascha.fiedler@med.uni-heidelberg.de
- First I would like you to better specify in the "methods" section how you have measured the gastric air content. This allows you to reproduce your study as faithfully as possible.
Response: Thank you to figure this out. We mentioned this problem in our discussion.
The evacuation of gastric air over the gastric tube by aspiration via a syringe is not as valid as gastric ultrasound. This is a limitation of our study and we need further randomized investigations with more patients to figure the risk of aspiration during ventilation with laryngeal masks out.
- I would like to see the individual contributions made by the authors in the section of the author's contribution. In fact from the draft it looks like you have pasted the template without compiling it.
Response: Thank you for pointing this out.
- Likewise, I would like to see the section on funding correctly filled out
Response: Study was financed by University Medical Centre Mannheim.
- Finally, I believe that you must also correctly fill in the "Acknowledgments" section.
Response: Thank you for your suggestion. We fill in the “acknowledgements” section.
Round 2
Reviewer 1 Report
Dear Authors,
I am sorry but I can not see relevant improvement of the manuscript concerning methodology, results and discussion of your study.
The primary and secondary outcome criteria, as postulated in the clinical trials registry DRKS00013254 is somewhat different to that you claim for in the actual manuscript.
Moreover, you changed your methodology post-hoc my reviewer comments. Face mask ventilation was part of the study protocol in the first version of the manuscript, now it is omitted.
I suggest you repeat the study with a sound protocol and valid methods.